# Synthesis of Green Recyclable Magnetic Iron Oxide Nanomaterials Coated by Hydrophobic Plant Extracts for Efficient Collection of Oil Spills

**DOI:** 10.3390/nano9101505

**Published:** 2019-10-22

**Authors:** Mahmood M. S. Abdullah, Ayman M. Atta, Hamad A. Al-Lohedan, Hamad Z. Alkhathlan, Merajuddin. Khan, Abdelrahman O. Ezzat

**Affiliations:** 1Surfactants Research Chair, Department of Chemistry, College of Science, King Saud University, P.O. Box 2455, Riyadh 11451, Saudi Arabiaao_ezzat@yahoo.com (A.O.E.); 2Department of Chemistry, Faculty of Applied Science, University of Taiz, P.O. Box: 4007, Taiz 009674, Yemen; 3Department of Chemistry, College of Science, King Saud University, P.O. Box 2455, Riyadh 11451, Saudi Arabiamkhan3@ksu.edu.sa (M.K.)

**Keywords:** magnetite nanomaterials, crude oil, oil spills, hydrophobic extracts

## Abstract

A facile method for synthesis of environmentally friendly magnetite nanomaterials (MNMs) was applied using hydrophobic biocomponents as capping and stabilizing agents. The biocomponents were extracted from *Matricaria aurea* (MAE) and *Ochradenus baccatus* (OBE) and used for the surface modification of MNMs to increase their dispersion efficiency on the collection of heavy crude oil spills. Synthesized MNM samples (MAE-MNMs and OBE-MNMs) were verified using thermogravimetric analysis; Fourier-transform infrared spectroscopy; transmission electron microscopy; dynamic light scattering, and vibrating-sample magnetometry. The application of these nanomaterials in the collection of oil spill showed that the MAE-MNMs and OBE-MNMs successfully collected 95% and 91% of the oil spill, respectively. These results support the potential use of these materials as eco-friendly composites for the successful collection of oil spills that might occur during offshore operations.

## 1. Introduction

One of the most known sources of marine pollution is oil spills produced from frequent accidents during the production and transportation of crude oil and its derivatives. Because conventional cleaning methods such as skimming and booming are considerably expensive, researchers are motivated to search for novel, cost effective methods [1,2]. Under this scope, a variety of chemicals, including polymers, surfactants and ionic liquids, are widely applied as oil spill dispersants, sorbents, and collectors [3,4,5,6,7,8,9,10,11,12,13]. Among the currently available methods, the oil spill collector’s technique has been the most applied in recent years due to its high efficiency, reusability, low cost and the ability to capture crude without dispersion into water [14,15,16]. In our previous studies, new synthesized magnetic nanomaterials (MNMs) used different natural capping agents such as asphaltene and plant extracts and applied as oil spill collectors [16,17,18]. It was noticed that the efficiency to collect heavy crude oil increased with using more hydrophobic MNMs [16,17,18]. 

According to the green chemistry principles, it is necessary to develop new substances alternative to the currently used chemicals in oil spill removals that contribute additionally to the marine pollution. New environmental regulations motivate researchers to consider alternative methods and chemicals that they are environmentally friendly. 

Recently, various natural products have extensively been utilized as capping, stabilizing, and reducing agents on nanomaterial synthesis due to their low cost, eco-friendly, availability, sustainability, and non-toxicity [19,20]. Many studies on the synthesis of MNMs used natural products such as, plant extracts [21,22,23,24,25], fungi [26,27,28], and biomolecules [29,30,31,32,33,34]. Plant parts, including leaves, fruits, and peels were also used successfully as stabilizing and capping agents in the preparation of MNMs [21,22,23,24,25]. *M. aurea* and *O. baccatus* plants belong to the Asteraceae and Resedaceae family, respectively. These plants are used extensively as medicinal plants for treatments of several diseases including hemorrhoids, colic, skin cracking, urinary tract infection, coughing, back pain, fistula, bacterial infections, and malaria [35,36,37]. They are fragrant and dioecious herbs widely growing in several regions of the world such as South Europe, North Africa, Middle East, and Asia [38]. The use of extracts from these plants for the synthesis of magnetite nanomaterials is being investigated for the first time in the present work. The aim of this work extended to utilize the n-hexane extracts of *M. aurea* and *O. baccatus* aerial parts as capping and stabilizing agents for the preparation of MNMs. The efficiency of the formed MNMs in the collection of oil spill was evaluated by taking advantage of the strong hydrophobicity of the capping agents.

## 2. Materials and Methods

### 2.1. Chemicals

All chemicals, i.e. as ferrous chloride tetrahydrate (FeCl_2_·4H_2_O ≥ 99%), ferric chloride hexahydrate (FeCl_3_·6H_2_O, 97%), ammonium hydroxide (25%), isopropanol and n-hexane were obtained from Aldrich Chemical Co. (Missouri, USA) and used without further purification. Saudi Arabia-based heavy crude oil was supplied from the Riyadh refinery unit: Aramco Co. (Riyadh, KSA) and the seawater was collected from the Arabian Gulf along the Saudi coast.

The aerial parts of *M. aurea* and *O. baccatus* were collected from the natural area of Rowdah Khuraim (Riyadh, KSA) during March 2016. A taxonomist in the herbarium division of King Saud University identified the species. The plant’s aerial parts were chopped into small pieces and air-dried under shade. After drying, the materials were separately soaked in n-hexane for 72 h thrice. The n-hexane extracts of *M. aurea* (MAE) and *O. baccatus* (OBE) were filtered and concentrated under reduced pressure using rotary evaporator.

### 2.2. Synthesis of Magnetite Nanomaterials (MNMs)

Capped MNMs were prepared by mixing solutions of 2:1 mole ratio ferric chloride hexahydrate to ferrous chloride tetrahydrate, (5.4 g Fe^3+^ and 2.0 g Fe^2+^ were dissolved in 100 mL of deionized water) with MAE or OBE solutions (2 g of each were dissolved separately in 50 mL of isopropanol). The mixture temperature was elevated to 50 °C under N_2_ atmosphere, to ensure the completion of the reaction; ammonia solution was added dropwise under continuous stirring for one hour, followed by adjustment of solutions pH to 10 with ammonium hydroxide. The formed MNMs were separated by the placement of external magnets, followed by sequential washing with isopropanol and water and then drying at ambient temperature. In this study, the synthesized MNMs coated by MAE or OBE are abbreviated as MAE-MNMs and OBE-MNMs, respectively.

### 2.3. Characterization of MNMs

The active functional groups in MAE and OBE extracts and MAE-MNMs and OBE-MNMs were investigated using Fourier-transform infrared spectroscopy (FTIR) (Thermo scientific, MN, USA). The experimental setup consisted of a Nexus 6700 FTIR spectrometer. The crystal lattice structure of the MNMs was confirmed by X-ray powder diffractometry (XRD) using a BDX-3300 diffractometer (Beijing University Equipment Manufacturer, Beijing, China) with a CuKa radiation source of wavelength λ = 1.5406 Å). The MNMs’ particles size in ethanol, their particle sizes, dispersion and zeta potential values were measured using dynamic light scattering (DLS) (Malvern Instruments, Malvern, UK). The experimental setup consisted of a Zetasizer 3000HS with a He-Ne laser source at wavelength λ = 633 nm. The surface morphology of the MAE-MNMs and OBE-MNMs was obtained using a JEOL JEM-2100F transmission electron microscope (TEM) (JEOL, Tokyo, Japan). The thermal stability of the MAE-MNMs and OBE-MNMs was measured by thermogravimetric analysis (TGA) using a Shimadzu DSC-60 thermal analyzer (Shimadzu Co, Canby, USA). The MNMs were heated under nitrogen atmosphere from 25 °C up to 700 °C at a heating rate of 10 °C/min.

Thin layers of MAE-MNMs or OBE-MNMs were deposited on the surface of a glass slide and their contact angle with seawater drops was determined using drop shape analysis (DSA) with a DSA-100 analyzer (Krüss GmbH, Hamburg, Germany). Prior to the measurements, small amounts of MA-MNMs or OB-MNMs were dispersed in ethanol solution followed by spreading onto the glass slide surface and then drying at 50 °C in a hot air oven. These steps were repeated three times until a thin film of MNMs was formed on the glass slide surface. The magnetic properties of MAE-MNMs and OBE-MNMs were obtained using vibrating-sample magnetometry (VSM) with a LDJ9600 magnetometer (LDJ Electronics, MI, USA) in a magnetic field of 20,000 Oe.

### 2.4. Efficiency of MAE-MNMs and OBE-MNMs as Oil Spill Collectors

About 1 mL of Saudi-based heavy crude oil was injected onto a surface of 250 mL seawater in a 500 mL beaker. Several ratios of the synthesized MNMs related to the oil contents were spread as solid powder to the crude oil and remixed using a glass rod for one minute. After 5 min, the adsorbed oil spill on the surface of MNMs was collected using external magnetic field, a permanent Nd-Fe-B magnet (4300 Gauss), and washed in a beaker with chloroform. The chloroform was evaporated under reduced pressure using rotary evaporator. The collected crude oil volume was determined as *V*_1_ and the original oil spill volume designated as *V*_0_. The residual oil on the surface of the brine water was extracted using chloroform. The ability of synthesized MNMs as oil spill collectors was calculated using Equation (1):
(1)$$C_{E}\;\left(\%\right)\;=\;\left(V_{0}/V_{1}\right)\;\times 100$$

For reusability purposes, after the heavy crude oil collection was finished the used MNMs samples were recollected using the permanent Nd-Fe-B magnet and washed several times with chloroform and ethanol solvents. Finally, the MNMs were air-dried and then reused for a new cycle of crude oil removal experiments. 

## 3. Results and Discussion

Before the extraction of polar active constituents from the raw *M. aurea* and *O. baccatus* plants materials, the hydrophobic constituents are usually defatted by n-hexane. During this process, large amounts of n-hexane extracts are obtained. Their hydrophobic properties such as the diversity of their hydrophobic constituents as well as their renewability, environmentally friendly nature and rapid extraction motivated us to apply these materials as capping agent during preparation of MNMs. Moreover, the synthesis of monodispersed MNMs in the presence of natural capping agents, extracted from *M. aurea* and *O. baccatus*, is one targets of the present work. Their efficiency in the synthesis of MNMs and their application towards the protection of marine species from the harmful presence of oil spills are another goal of this work. The natural hydrophobic extracts were selected as capping and stabilizing agents because the enhanced dispersity of the synthesized MNMs in crude oil and their efficiency in the collection of oil spill. 

### 3.1. Characterization of the Prepared MNMs 

The active functional groups of MAE, OBE and MAE-MNMs and OBE-MNMs are elucidated using FTIR spectra represented in Figure 1a–d. The spectra of MAE and OBE (Figure 1a,b) show strong absorption bands at 3447, 3420 cm^−1^; 2918, 2970 cm^−1^; 2850, 2932 cm^−1^; 1736, 1766 cm^−1^; and 1629, 1655 cm^−^1 corresponding to O-H, saturated C-H, C=O, and C=C stretching vibrations. The appearance of new bands of the MNMs spectra at 580 and 583 cm^−1^ (Figure 1c,d) correspond to the stretching vibrations of Fe-O, which confirms the formation of iron oxide as reported earlier [39].

The XRD diffraction patterns of the MAE-MNMs and OBE-MNMs are shown in Figure 2a,b. The appearance of characteristic peaks at 2θ angles 30.14°, 35.91°, 43.21°, 53.73°, 57.27°, 62.51° and 74.79° is associated with the indices 220, 311, 400, 422, 511, 440 and 622, respectively, as indicated in the Figure 2. Compared to the standard peaks [40], it is elucidated that the chemical structure of MNMs is unaffected by the surface modification using the phytoconstituents of MAE and OBE extracts. Additionally, the appearance of a broad peak at 19.11° corresponds to the presence of the extracts and confirms the successful coating of the MNMs [16].

The thermal stability of the formed MNMs was determined using TGA analysis (Figure 3). The initial weight loss of the MAE-MNMs and OBE-MNMs (at temperature increase up to 200 °C) is generally related to the evaporation of physio-adsorbed water or other solvents during the purification process. The degradation of the MAE-MNMs and the OBE-MNMs begins approximately at 200 °C and stops around 500 °C with a total weight loss of approximately 20% and 21%, respectively. Such material loss may be attributed to the decomposition of the MAE and OBE constituents. The magnetite content of the MAE-MNMs and the OBE-MNMs at 700 °C was measured to be 77% and 74%, respectively, suggesting an increase in the total amount of capping agent in the OBE-MNM samples. 

The TEM micrographs in Figure 4a,b show the surface morphology of the MAE-MNMs and the OBE-MNMs. The formation of irregular spherical structures with an average diameter of 9.6 ± 3 nm is observed. The thermograms exhibited limited variation in the particles size of the nanomaterials and this variation may be related to the diversity of the MAE and OBE phytoconstituents. The formed MNMs are smaller than those synthesized using extracts from different plant species [20,23,39]. Furthermore, Figure 4a shows that the MAE-MNMs agglomerate because the increase in the content of the MNMs forces them to aggregate. This was also observed at the TGA analysis due to their magnetic nature (Figure 3). 

DLS measurements of the MAE-MNMs and OBE-MNMs in an ethanol solvent revealed the particle size, dispersity and zeta potential values, as shown in Figure 5 and Figure 6, respectively. The average particle size and polydispersity index were 338.5 nm and 0.234, respectively for the MAE-MNMs; and 367.8 nm and 0.228, respectively, for the OBE-MNMs. The increase in particle diameter suggests agglomeration of the MNMs in ethanol, which showed considerable difference with the TEM results of the dried MNMs particles. The zeta potential measurements of the MAE-MNMs and OBE-MNMs (Figure 6a,b)) showed positive values at 8.35 and 17.1 mV, respectively. The higher zeta potential value of the OBE-MNMs confirms their higher colloidal stability in ethanol solvent, compared to MAE-MNMs. Notably, the positive values of zeta potential suggest an enhanced interaction between these MNMs with asphaltene in heavy crude oil, which has a negative surface charge during oil spill removal [41,42].

In our previous work [17,18], MNNs were capped by using hexane and chloroform extracts. The MNNs capped by hexane extracts showed higher hydrophobicity and dispersity in nonpolar organic solvents than that capped with chloroform extracts. For these reasons, the present work used either *M. aurea* or *O. baccatus* hexane extracts for capping the MNNs. Generally, the capped MNMs ability to collect oil spill is mainly affected by their ability to disperse in crude oil. So, an increase in the hydrophobicity of the capping agents, provided by the coated plant extracts, directly increases the MNMs’ dispersion efficiency in crude oil which further enhances their ability to collect oil spills. We found that the capped MNMs exhibited high dispersion in toluene, chloroform, xylene, and other low polar organic solvents, but in seawater the dispersion was very low. The hydrophobicity of the synthesized MNMs was determined by contact angle measurements. Figure 7 shows the contact angles of the MAE-MNMs and OBE-MNMs at 128° and 111°, respectively. This observation corresponds to an increased hydrophobicity of the MAE as a capping agent compared to the OBE. Moreover, the appearance of absorption bands in FTIR spectra (Figure 1c and d) between 3000–2850 cm^−1^ related to CH stretching vibrations confirms the presence of hydrophobic aliphatic chains in these extracts on the MNMs surfaces.

The ability of the prepared MNMs to response for an external magnetic field was evaluated via measuring magnetic properties such as saturation magnetization (Ms), remanent magnetization (Mr) and coercivity (Hc) by VSM magnetic hysteresis loops at 25 °C. Figure 8 elucidates the magnetization of both MAE-MNMs and OBE-MNMs was not saturated even at highest field of (20,000 Oe or 2 T). This can be referred the capping of MAE or OBE on the magnetite surfaces leads to form non-crystalline phase which is responsible for the non-saturation of the hysteresis loop (Figure 8) even at high fields which is termed as high field susceptibility [43]. It indicates the formation of magnetically hard components that can be associated to surface spin disorder [43]. Figure 8 shows absence of hysteresis loops, remanence, or coercivity suggesting that the MAE-MNMs and OBE-MNMs are superparamagnetic materials. At 20,000 Oe, magnetization values of OBE-MNMs and MAE-MNMs are determined to be 55.03 and 48.93 emu/g, respectively, which is enough for the separation of the MNMs by the external magnetic field. Additionally, the increased magnetization value of the OBE-MNMs corresponds to lower amount of capping agents (higher magnetic content) as compared to the MAE-MNMs. This observation was also confirmed by the TGA analysis (Figure 3).

### 3.2. Efficiency of MAE-MNMs and OBE-MNMs As Oil Spill Collectors

Increase the hydrophobicity of the prepared MNMs and their supermagnetic nature make them suitable candidates in oil spill remediation. Therefore, we applied several ratios of MNM to crude oil (1:1 up to 1:50) and evaluated their efficiency on the collection of Arabian heavy crude oil. The results, presented in Table 1, indicate that the efficiency of the MNMs in collecting oil spills increases with the increase of their ratio to crude oil. The best ratio of the MAE-MNMs and the OBE-MNMs was 1:10 with removal of 92% and 88% of oil spills, respectively. These data also showed a high efficiency of the MNMs to crude oil ratio at 1:1, which can be attributed to the aggregation of the MNMs that effects the attraction between these nanoparticles when an external magnetic field is applied. 

The MNM samples were re-used five times. Each time samples were separated using an external magnet and washed thrice with chloroform followed by an ethanol, air-drying, and direct utilization for the next cycle. A ratio of MNMs to crude oil of 1:10 was chosen to evaluate the ability of the recovered MNMs in the collection of oil spill and their results are represented in Figure 9. It was found that slightly decreasing in their efficiency with an increasing in the number of the reused cycles. This observation can be attributed to increase agglomeration of MNMs and adsorption of some crude oil components such as asphaltene on the MNMs surfaces with each additional cycle. The agglomeration of MNMs and adsorption of the residual crude oil components on their surfaces were confirmed by carrying out both particle sizes measurements and TGA analysis as showed in Figure 5b,d and Figure 10a,b, respectively. It was observed that, the MNMs particles agglomerated after 5 cycles to reflect the adsorption and interaction of crude oil components on their surfaces. TGA thermograms (Figure 10a,b) elucidate that the amount of the remained residual metal oxide above 650 °C decreased with increasing in the number of cycles. Moreover, the TGA thermograms (Figure 10a,b) confirm the increasing of the amount of organics adsorbed molecules on the MNMs surfaces after five cycles when compared with the original MNMs. 

## 4. Conclusions

In this study we used two different plants extracts obtained from *M. aurea* and *O. baccatus* to synthesize capped hydrophobic magnetite nanomaterials. It was found that the diversity of the biocomponents in both extracts decreased the particle size diameter of the prepared MNMs and increased their ability to disperse and interact with the constituents of crude oil. The incorporation of the biocomponents of MAE and OBE to the MNMs and the existence of these nanomaterials were confirmed by FTIR and powdered XRD analyses. The contact angles measurements demonstrated an increase in the hydrophobicity of the prepared MNMs, suggesting an increase in their ability to disperse in crude oil compared to water media. The DLS and TEM measurements revealed the microstructural form of the nanomaterials with an average diameter of 9.6 ± 3 nm. The synthesized MNMs showed a supermagnetic behavior and the magnetization value was lower for OBE-MNMs compared to MAE-MNMs. This increase suggests an increase in the content of the MNMs. Furthermore, the MAE-MNMs showed a higher ability for collecting of oil spills because of their ability to disperse in crude oil more effectively compared to the OBE-MNMs. The reusability of the MNMs was demonstrated for at least five cycles, displaying a slight loss in the efficiency with every repeating cycle. To conclude, the availability of the *M. aurea* and *O. baccatus* plants in nature, their environmentally friendly extracted hydrophobic biocomponents, and the simplicity of MNM synthesis all make these nanomaterials suitable for the remediation of oil spills occurring during offshore operations. 

## Figures and Tables

**Figure 1 nanomaterials-09-01505-f001:**
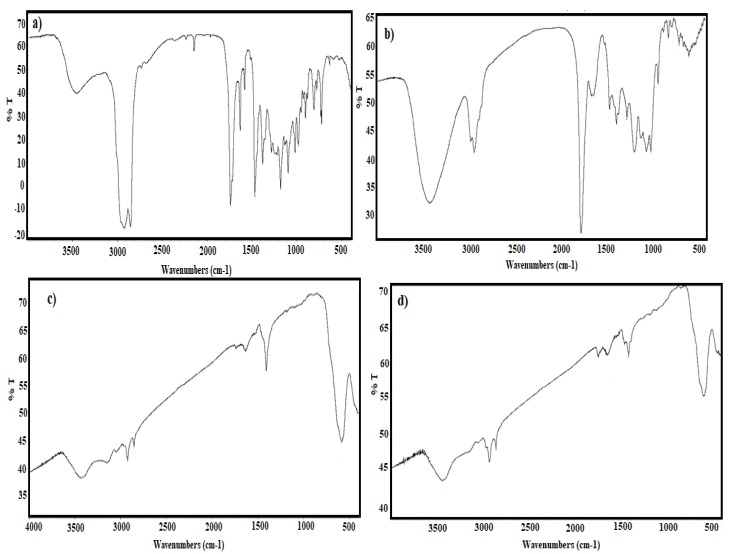
FTIR spectra of (**a**) M. aurea extract (MAE), (**b**) O. baccatus extract (OBE), (**c**) magnetite nanomaterials coated by MAE (MAE-MNMs) and (**d**) magnetite nanomaterials coated by OBE (OBE-MNMs).

**Figure 2 nanomaterials-09-01505-f002:**
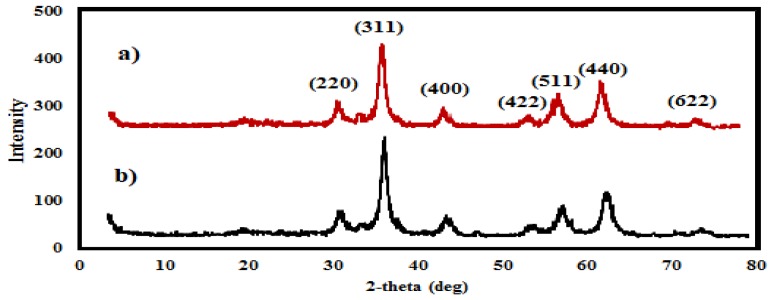
Powdered XRD diffraction patterns of magnetite nanomaterials coated by (**a**) *M. aurea* extract (MAE-MNMs) and (**b**) *O. baccatus* extract (OBE-MNMs). The peaks indices are indicated.

**Figure 3 nanomaterials-09-01505-f003:**
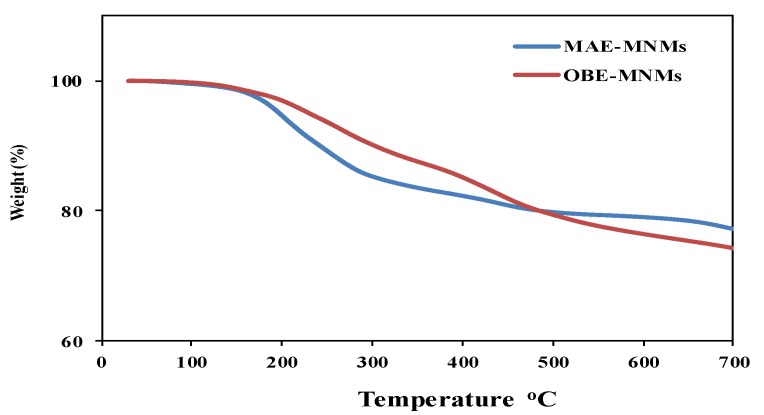
TGA thermograms of magnetite nanomaterials coated by *M. aurea* extract (MAE-MNMs) (blue curve) and by *O. baccatus* extract (OBE-MNMs) (red curve).

**Figure 4 nanomaterials-09-01505-f004:**
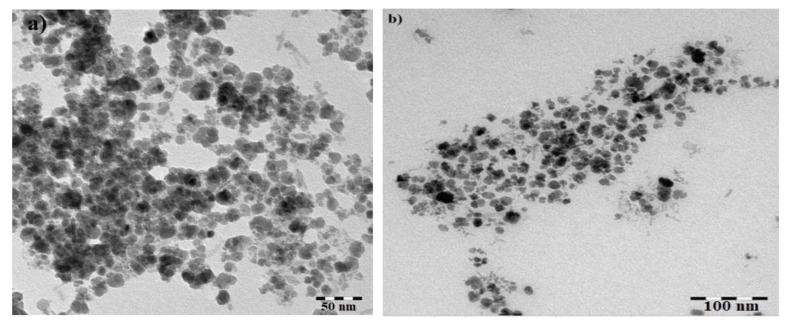
TEM micrographs of magnetite nanomaterials coated by (**a**) *M. aurea* extract (MAE-MNMs) and (**b**) *O. baccatus* extract (OBE-MNMs).

**Figure 5 nanomaterials-09-01505-f005:**
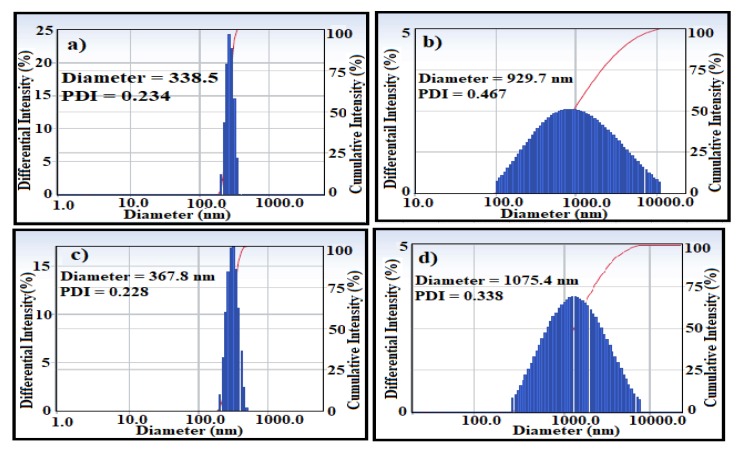
Particle size distribution of magnetite nanomaterials coated by (**a**) *M. aurea* extract (MAE-MNMs), (**b**) *M. aurea* extract (MAE-MNMs) after 5 cycles, (**c**) *O. baccatus* extract (OBE-MNMs), and (**d**) *O. baccatus* extract (OBE-MNMs) after 5 cycles in ethanol solvent.

**Figure 6 nanomaterials-09-01505-f006:**
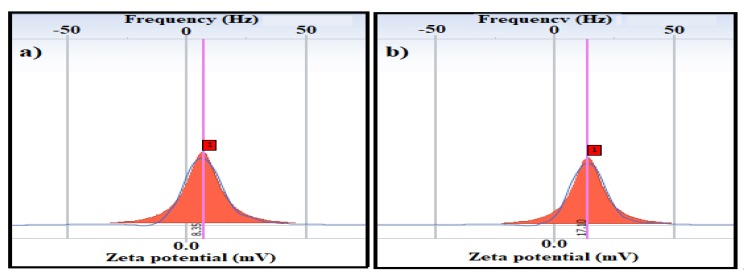
Zeta potential of magnetite nanomaterials coated by (**a**) *M. aurea* extract (MAE-MNMs) and (**b**) *O. baccatus* extract (OBE-MNMs) in ethanol.

**Figure 7 nanomaterials-09-01505-f007:**
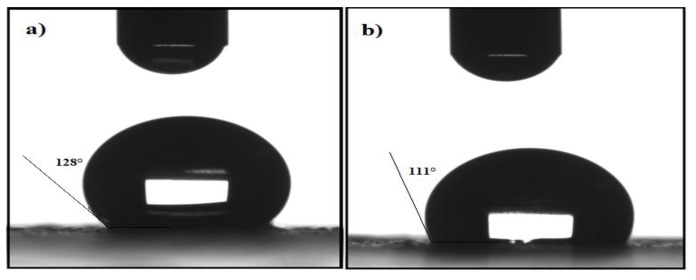
Contact angles of water on magnetite nanomaterials coated by (**a**) M. aurea extract (MAE-MNMs) and (**b**) *O. baccatus* extract (OBE-MNMs).

**Figure 8 nanomaterials-09-01505-f008:**
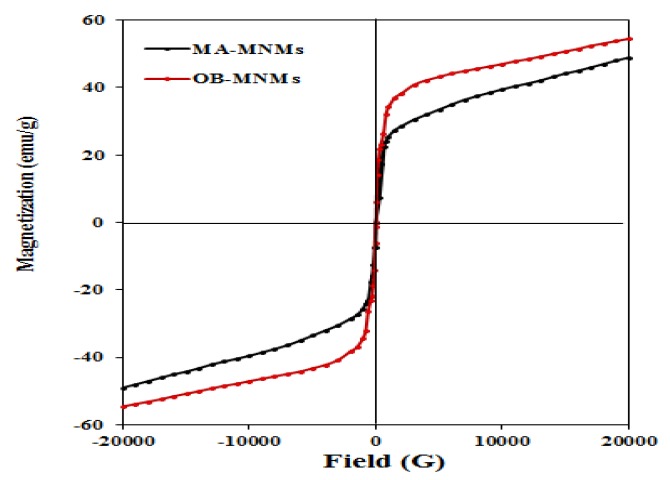
VSM loop of magnetite nanomaterials coated by (**a**) *M. aurea* extract (MAE-MNMs) (black curve) and (**b**) *O. baccatus* extract (OBE-MNMs) (red curve) at 298 K.

**Figure 9 nanomaterials-09-01505-f009:**
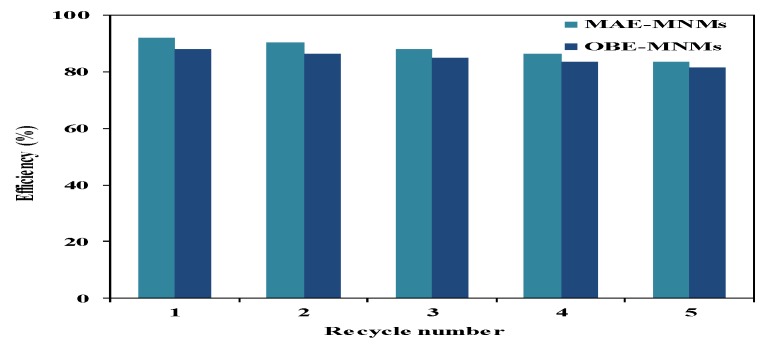
Efficiency on the collection of oil spills of recycled magnetite nanomaterials coated by *M. aurea* extract (MAE-MNMs) (light blue bars) and *O. baccatus extract* (OBE-MNMs) (dark blue bars).

**Figure 10 nanomaterials-09-01505-f010:**
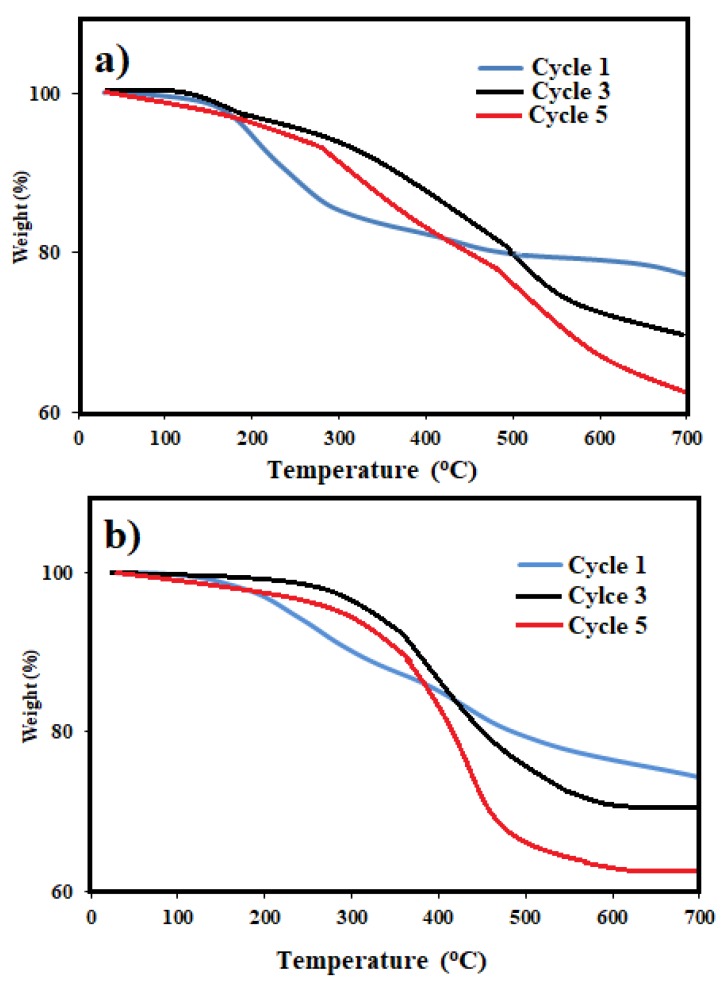
TGA thermograms of magnetite nanomaterials (**a**) coated by M. aurea extract (MAE-MNMs) and (**b**) by O. baccatus extract (OBE-MNMs) after different cycles.

**Table 1 nanomaterials-09-01505-t001:** Efficiency (%) of synthesized magnetite nanomaterials (MNMs) on the collection of oil spills at different ratios of MNMs to crude oil.

MNMs	Collection Efficiency (%) Using Different (MNMs: Crude Oil) ) (Wt.:Vol %)			
	1:1	1:10	1:25	1:50
MAE-MNM	95	92	88	80
OBE-MNM	91	88	83	76

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
