# Peer review of "Synthesis of Green Recyclable Magnetic Iron Oxide Nanomaterials Coated by Hydrophobic Plant Extracts for Efficient Collection of Oil Spills"

_nanomaterials, 2019, doi:10.3390/nano9101505_

Round 1

Reviewer 1 Report

This report by Abdullah et al. describes the preparation, characterization and testing of coated iron oxide(magnetite) as a material to treat oil spills.   The authors use extracts from two different plant species to create a hydrophobic coating on the nanoparticles. These absorb(adsorb?) the oil with separation facilitated by the magnetic properties.  FTIR, DLS, XRD, TGA, TEM and contact angle measurements of the product are carried out.  The modified magnetite product is added in various ratios to crude and extraction efficiency found.

Magnetic separations have been used for many years, though this approach with nanoparticles for oil cleanup is novel.   The research in this paper however continues along the lines of prior publications by this group. It is in fact quite similar to reference 17, so that it is appropriate for the authors to contrast the present submission with their prior publication.

There is very little text in the Results section so that restructuring of the paper into a Results and Discussion section is recommended for the benefit of the reader.  Many typographical errors appear throughout the narrative and the writing could be improved.  This includes proper form for genus and species which is incorrectly shown throughout.  Proper form for writing genus is capitalized, ordinary type(e.g. Ochradenus, see line 17).  Species designation should be low case and italics(e.g. baccatus, see line 49).

To measure efficiency of the modified magnetite particles as an oil spill collector, the experimentalist poured heavy crude on top of sea water in a beaker and then stirred for 1 minute.  Is it realistic in actual application to stir a large oil spill in open water?   What is the result for recovery if the MNMs are applied without stirring?  Adding a test like this would more likely reflect use of the technology in the field.

The oil recovered in the experiments is not quite clear(Section 2.4).  MNMs with oil were collected using a permanent magnet and then residual oil was extracted with chloroform.  Does Vo include both of these steps?    What was the total oil recovered after both steps? 

The use of chloroform in an actual spill would not be permissible in many places.  Did the authors consider a second application of MNMs to remove residual oil?

Coatings do not seem to be covalently bonded to the surface, rather they are physisorbed on the MNMs.   The coatings are thick as shown by the TEM and TGA data.  How much of this coating is lost with a cycle of oil collection and rinsing with organic solvents?  It would be interesting to see the results of TGA after several cycles.

Contact angles of 128 o and 111o of this magnitude occur as a result of surface topography and measurement technique used.   Findings may not reflect the affinity of the dispersed nanoparticles for the oil.   The authors are encouraged to consider their FTIR data that might help to reinforce their interpretation.

One does not weigh out iron ions. Provide the mass of the reagents in Section 2.2.

Specify in Table 1 whether mass or volume ratio has been used. 

Is there evidence to support the description that these particles are “environmentally friendly”? The plants might be described as eco-friendly, but that may not be the case for extracted species. Use of the plants for medicinal purposes indicates that derived compounds are bioactive. 

There have been concerns about toxicity of some nanoparticles.   Is there data supporting the safety of magnetite nanoparticles?

The scales in Figure 4 are not legible.  In the narrative the size of both MNMs are given at 9.3 nm, but the MAE-MNM magnetite nanoparticles look to be larger.   Explain.

Line

33: Use “as” instead of “in”                                                                                                                

34:  oil spill collector technique has been

35: low cost and the ability to capture crude

38: extracts. We noticed increased efficiency to collect heavy crude oil with more hydrophobic MSMs[16-18].

54: the use of extracts from these plants for the synthesis of

61-62:  Subscript 2 and 3 in ferrous and ferric chloride formulas.

75: Capped MNMs were prepared by mixing

80: by adjustment of solutions pH to 10 with ammonium hydroxide.

114: brine

159: polydispersity

197: 2970 cm-1 and 2850. There are 5 sets of wave numbers for 4 functional group assignments so that “respectively” does not apply.

200: Bands for Fe-O does not indicate purity.  It only indicates an iron oxide is present.

229: which showed considerable difference

232:  What is the higher zeta potential confirming?  This is not clear.

Author Response

Reviewer 1

This report by Abdullah et al. describes the preparation, characterization and testing of coated iron oxide(magnetite) as a material to treat oil spills. The authors use extracts from two different plant species to create a hydrophobic coating on the nanoparticles. These absorb(adsorb?) the oil with separation facilitated by the magnetic properties.  FTIR, DLS, XRD, TGA, TEM and contact angle measurements of the product are carried out.  The modified magnetite product is added in various ratios to crude and extraction efficiency found. Magnetic separations have been used for many years, though this approach with nanoparticles for oil cleanup is novel. The research in this paper however continues along the lines of prior publications by this group. It is in fact quite similar to reference 17, so that it is appropriate for the authors to contrast the present submission with their prior publication.

Answer

Abbreviated discussion was added to the manuscript to compare the prior work with the current one.

There is very little text in the Results section so that restructuring of the paper into a Results and Discussion section is recommended for the benefit of the reader. Many typographical errors appear throughout the narrative and the writing could be improved.  This includes proper form for genus and species which is incorrectly shown throughout.  Proper form for writing genus is capitalized, ordinary type(e.g. Ochradenus, see line 17).  Species designation should be low case and italics(e.g. baccatus, see line 49).

Answer

Results and discussion were emerged in one section according to the reviewer comment. Typo errors were corrected in whole manuscript. In the first time (line 17) the full names of plants were written including genus and species, in the rest of manuscript we abbreviated their name as referenced in many studies. In line 49 the names of plants were modified to be M. aurea and O. baccatus

To measure efficiency of the modified magnetite particles as an oil spill collector, the experimentalist poured heavy crude on top of sea water in a beaker and then stirred for 1 minute. Is it realistic in actual application to stir a large oil spill in open water?   What is the result for recovery if the MNMs are applied without stirring?  Adding a test like this would more likely reflect use of the technology in the field.

Answer

Waves in oceans and seas play very important role for mixing and homogenizing these nanomaterials with crude oil. Most of the previous studies reported the mixing of MNMs with crude oil using glass rode or stirring using magnetic stirrer to simulate the waves action in oceans and seas. In this work we preferred to use glass rode over magnetic bar due to attractions between MNMs and magnetic bar that disturbs the magnetic strength of MNMs.

The oil recovered in the experiments is not quite clear (Section 2.4). MNMs with oil were collected using a permanent magnet and then residual oil was extracted with chloroform.  Does Vo include both of these steps?    What was the total oil recovered after both steps?

Answer

The paragraph was modified to clarify these procedures to be “the adsorbed oil spill on the surface of MNMs was collected using external magnetic field (a permanent Nd-Fe-B magnet (4300 Gauss)) and washed in a beaker with chloroform, followed by evaporation of chloroform under reduced pressure using rotary evaporator to obtain the collected crude oil. The residual oil on the surface of the brine water was extracted using chloroform.”.

The use of chloroform in an actual spill would not be permissible in many places. Did the authors consider a second application of MNMs to remove residual oil?

Answer: Chloroform used to dilute the crude oil and to reduce its viscosity to easily separate the MNMs. Chloroform volatile organic solvent that can be easily removed by rotary evaporator. It is very important to recycle the crude oil without MNMs, chloroform without crude oil and MNMs or recycling of MNMs to collect more crude oil.

Coatings do not seem to be covalently bonded to the surface, rather they are physisorbed on the MNMs. The coatings are thick as shown by the TEM and TGA data.  How much of this coating is lost with a cycle of oil collection and rinsing with organic solvents?  It would be interesting to see the results of TGA after several cycles.

Answer

TGA analysis was carried out after several cycles and added to the manuscript. TGA data indicated that the amount of capping increased with increase the number of cycle (although we washed them several times with chloroform followed by ethanol and dried properly) which could be attributed to the adsorption of some crude oil components on the surface of MNMs. To interpret the slightly decreasing in MNMs efficiency with an increase in the number of cycles, we measured the Particle sizes of the recycled MNMs after 5 cycles we found an increase in their particles size which indicates increase agglomeration of MNMs that leads to decrease their dispersion in crude oil and lack of their magnetic strength for collecting crude oil.    

Contact angles of 128° and 111° of this magnitude occur as a result of surface topography and measurement technique used. Findings may not reflect the affinity of the dispersed nanoparticles for the oil.   The authors are encouraged to consider their FTIR data that might help to reinforce their interpretation.

Answer

FTIR data were discussed to interpret the hydrophobicity of the formed MNMs, although, there are many studies focused on the use of contact angles measurements as indicator to evaluate the hydrophobicity of layers on glass surface or steel surface.

One does not weigh out iron ions. Provide the mass of the reagents in Section 2.2.

Answer

The mass of all reagents already exists

Specify in Table 1 whether mass or volume ratio has been used.

Answer

The ratio was specified and added to manuscript

Is there evidence to support the description that these particles are “environmentally friendly”? The plants might be described as eco-friendly, but that may not be the case for extracted species. Use of the plants for medicinal purposes indicates that derived compounds are bioactive.

Answer

We didn’t do any antimicrobial studies, although, many studies reported use of the extracts and bio active isolated compounds from these plants as antimicrobial and for treatment of different diseases. These plants are used widely in folk medicine.  In many countries, they use M. aurea as tea. The following references mentioned to the medical applications of these plants extracts: 

Antimicrobial Activity and Chemical Composition of Flowers of Matricaria aurea a Native Herb of Saudi Arabia. International Journal of Pharmacology, Volume 12 (6): 576-586, 2016. Antimicrobial Activity of Polyphenols and Alkaloids in Middle Eastern Plants. Frontiers in Microbiology, May 2019 | Volume 10 | Article 911, doi: 10.3389/fmicb.2019.00911 Anti-inflammatory activity, safety and protective effects of Leptadenia pyrotechnica, Haloxylon salicornicum and Ochradenus baccatus in ulcerative colitis. Phytopharmacology 2012, 2(1) 58-71. Antimicrobial Activity of Extracts of some Plants Collected from the Kingdom of Saudi Arabia. Journal of King Abdulaziz University - Medical Sciences, Vol. 15 No. 1, pp: 25-33 (2008). There have been concerns about toxicity of some nanoparticles. Is there data supporting the safety of magnetite nanoparticles?

Answer

There are many papers reported safety, low toxicity and medical applications of magnetite nanoparticles. Such as:

1- Medical Application of Functionalized Magnetic Nanoparticles, Journal of Bioscience and Bioengineering, Volume 100, Issue 1, July 2005, Pages 1-11

2- Potential toxicity of superparamagnetic iron oxide nanoparticles (SPION), Nano Review. 2010, doi: 10.3402/nano.v1i0.5358

3- Dextran and albumin derivatised iron oxide nanoparticles: influence on fibroblasts in vitro. Biomaterials. 2003 Nov;24(25):4551-7.

4- Synthesis and surface engineering of iron oxide nanoparticles for biomedical applications. Biomaterials Volume 26, Issue 18, June 2005, Pages 3995-4021

5- Nanoparticle iron chelators: a new therapeutic approach in Alzheimer disease and other neurologic disorders associated with trace metal imbalance. Neuroscience Letters 2006 Oct 9;406(3):189-93.

The scales in Figure 4 are not legible. In the narrative the size of both MNMs are given at 9.3 nm, but the MAE-MNM magnetite nanoparticles look to be larger.  

Answer

The scale was clarified. In MAE-MNM the scale 50 nm, and the value 9.3 nm was average diameter, although there are some agglomerations their sizes bigger than this value.

Line 33: Use “as” instead of “in”

Answer

“as” was used instead of “in”

34: oil spill collector technique has been

Answer

The text was modified to be “oil spill collector’s technique has been”

35: low cost and the ability to capture crude

Answer

The text was modified to be “low cost and the ability to capture crude”

38: extracts. We noticed increased efficiency to collect heavy crude oil with more hydrophobic MSMs [16-18].

Answer

The sentence was modified to be “We noticed increased efficiency to collect heavy crude oil with more hydrophobic MSMs”

54: the use of extracts from these plants for the synthesis of

Answer

The sentence was modified to be “We report the use of extracts from these plants for the synthesis of magnetite nanomaterials, which is being investigated for the first time.”

61-62: Subscript 2 and 3 in ferrous and ferric chloride formulas.

Answer

Ferrous and ferric chloride formulas were corrected.

75: Capped MNMs were prepared by mixing

Answer

The sentence was modified to be “Capped MNMs were prepared by mixing solutions of 2:1 M ferric chloride hexahydrate to ferrous chloride tetrahydrate, (5.4 g Fe3+ and 2.0 g Fe2+ were dissolved in 100 mL of deionized water) with MAE or OBE solutions (2 g of each were dissolved separately in 50 mL of isopropanol).”

80: by adjustment of solutions pH to 10 with ammonium hydroxide. {Correct?]

Answer

The sentence was modified.

114: brine

Answer

The typo error was corrected

159: polydispersity

Answer

The typo error was corrected

197: 2970 cm-1 and 2850, There are 5 sets of wave numbers for 4 functional group assignments so that “respectively” does not apply.

Answer

The aliphatic C-H has two starching bands, for this reason you saw 5 sets of wave numbers for 4 functional groups, so we removed respectively from the text.

200: Bands for Fe-O does not indicate purity. It only indicates an iron oxide is present.

Answer

The sentence was modified to be “The appearance of new bands of the MNMs spectra at 580 and 583 cm-1 (Figure 1c and d) correspond to the stretching vibrations of Fe-O, which confirms the formation of iron oxide as reported earlier”.

229: which showed considerable difference

Answer

The sentence was modified to be “The increase in particle diameter suggests agglomeration of the MNMs in ethanol, which showed considerable difference with the TEM results of the dried MNMs particles.”

232: What is the higher zeta potential confirming?  This is not clear.

Answer

Increase the value of zeta potential indicates the colloidal stability of the dispersed nanoparticles in solution. The zeta potential value of OBE-MNMs is higher than that MAE-MNMs, this means it has higher stability in ethanol than that MAE-MNMs. We added colloidal stability to clarify the meaning of sentence.

Reviewer 2 Report

The authors describe a study obtaining plant extracts using hexane extraction and concentration, followed by reduction of iron compounds to produce magnetite nanoparticles coated with organic plant extract molecules. They characterize the magnetic nanomaterials obtained through said process and their application in oil removal from water. The manuscript can be suitable for publication in MDPI nanomaterials after the following changes:

Figure1: Make the axis labels larger (Wavenumber and %T and the numbers on the axes should be much larger to enhance readability)

Figure 2: Higher resolution graphic is needed.

Figure 4: They need to state in the Figure caption how large the scale bar is.

Figure 5: Figure resolution is too low; reading the axis labels is almost impossible. Increase font size on all axis labels, increase font size on axis numbers,

Figure 6: Increase figure resolution and font sizes for all axes.

Figure 7: needs scale bars inside the images

Figure 9: y-axis label has a typographical error: It should be “efficiency” not “effeciency”

Line 47: When they reference magnetic nanoparticles synthesized with biomolecules, the following 4 references are highly relevant and should be added to the references list: Nano Letters, 17(12): 7932-7939, 2017; ACS Nano, 6(8): 6776–6785, 2012; Nature Methods vol. 9, p. 1113–1119 (2012); Journal of the Taiwan Institute of Chemical Engineers Vol. 97, April 2019, Pages 227-236. These papers describe magnetic separation of target biomolecules and cells using smart protein coatings, as well as the application of magnetic nanoparticles to oil spill clean up. As such they are highly relevant and should be cited.

Line 193: change “Van deer Waals” to “van der Waals”

Author Response

Reviewer 2

The authors describe a study obtaining plant extracts using hexane extraction and concentration, followed by reduction of iron compounds to produce magnetite nanoparticles coated with organic plant extract molecules. They characterize the magnetic nanomaterials obtained through said process and their application in oil removal from water. The manuscript can be suitable for publication in MDPI nanomaterials after the following changes:

Figure1: Make the axis labels larger (Wavenumber and %T and the numbers on the axes should be much larger to enhance readability)

Answer:

The axis labels were rewritten to be easy for reading.

Figure 2: Higher resolution graphic is needed.

Answer:

The resolution of this figure was improved.

Figure 4: They need to state in the Figure caption how large the scale bar is.

Answer:

The font of figure caption was rewritten clearly.

Figure 5: Figure resolution is too low; reading the axis labels is almost impossible. Increase font size on all axis labels, increase font size on axis numbers,

Answer:

The resolution of this figure was improved and font sizes was increased

Figure 6: Increase figure resolution and font sizes for all axes.

Answer:

The resolution and font sizes of this figure were improved

Figure 7: needs scale bars inside the images

Answer:

These images were taken from instrument; the value of contact angle appears in software or it can be measured manually. We added the values of contact angles to these images.

Figure 9: y-axis label has a typographical error: It should be “efficiency” not “effeciency”

Answer:

The typo error was corrected.

Line 47: When they reference magnetic nanoparticles synthesized with biomolecules, the following 4 references are highly relevant and should be added to the references list: Nano Letters, 17(12): 7932-7939, 2017; ACS Nano, 6(8): 6776–6785, 2012; Nature Methods vol. 9, p. 1113–1119 (2012); Journal of the Taiwan Institute of Chemical Engineers Vol. 97, April 2019, Pages 227-236. These papers describe magnetic separation of target biomolecules and cells using smart protein coatings, as well as the application of magnetic nanoparticles to oil spill clean up. As such they are highly relevant and should be cited.

Answer:

All of these references were added to the manuscript according to reviewer comments.

Line 193: change “Van deer Waals” to “van der Waals”

Answer:

This paragraph was removed due to emerging results and discussion as the another reviewer suggest.